# Financial Toxicity in Swiss Cancer Patients Treated with Proton Therapy: An Observational Cross-Sectional Study on Self-Reported Outcome

**DOI:** 10.3390/cancers15235498

**Published:** 2023-11-21

**Authors:** Barbara Bachtiary, Leonie Grawehr, Filippo Grillo Ruggieri, Ulrike Held, Damien C. Weber

**Affiliations:** 1Centre for Proton Therapy, Paul Scherrer Institute, ETH Domain, 5232 Villigen, Switzerland; f.grilloruggieri@gmail.com (F.G.R.); damien.weber@psi.ch (D.C.W.); 2Faculty of Medicine, University of Zurich, 8006 Zurich, Switzerland; leonie.grawehr@uzh.ch; 3Department of Biostatistics at Epidemiology, Biostatistics and Prevention Institute, University of Zurich, 8001 Zurich, Switzerland; ulrike.held@uzh.ch; 4Department of Radiation Oncology, Inselspital, Bern University Hospital, University of Bern, 3010 Bern, Switzerland; 5Department of Radiation Oncology, University Hospital of Zurich, University of Zurich, 8091 Zurich, Switzerland

**Keywords:** proton therapy, financial toxicity, financial burden, out-of-pocket costs

## Abstract

**Simple Summary:**

Proton therapy is a highly effective type of radiotherapy that is used to treat cancer while minimizing damage to healthy tissue. In Switzerland, health insurance typically covers the cost of proton therapy. However, patients may still need to pay for their lodging and travel expenses during the treatment’s 5–7 week duration. This can lead to financial stress, also known as financial toxicity. The present study is the first to measure patient-reported financial toxicity in a group of patients receiving proton therapy, including both patients and caregivers of pediatric cancer patients. It shows that many patients receiving proton therapy experience financial toxicity, which is measured by the COST score. Low income, long travel distances, and marriage can contribute to this financial burden. Patients often had to save money by cutting back on expenses in different areas. The study emphasizes the importance of considering financial aspects as part of cancer patients’ overall care during proton therapy.

**Abstract:**

Background: Proton therapy is indicated for cancers that would be difficult to treat with conventional radiotherapy. Compulsory healthcare insurance covers the costs of this therapy in Switzerland, but this does not mean that proton therapy is cost-neutral for every cancer patient. Significant out-of-pocket (OOP) costs may arise due to expenses associated with proton therapy, and patients may experience treatment-related financial distress—an effect known as “financial toxicity.” This study investigates the financial toxicity of patients undergoing proton therapy in a high-income country with a compulsory health insurance policy. Methods: Between September 2019 and November 2021, 146 Swiss cancer patients treated with proton therapy participated in this study, of whom 90 (62%) were adults and 56 (38%) were caregivers of child cancer patients. Financial toxicity was assessed using the FACIT Comprehensive Score for Financial Toxicity (COST). OOP costs during proton therapy were recorded weekly, and financial coping strategies were captured at the end of treatment. Findings: The median COST score, indicating financial toxicity, was 29.9 (IQR 21.0; 36.0) for all patients, 30.0 (IQR 21.3; 37.9) for adults, and 28.0 (IQR 20.5; 34.0) for children’s caregivers. Higher income (estimate 8.1, 95% CI 3.7 to 12.4, *p* ≤ 0.001) was significantly associated with higher COST scores, indicating less financial toxicity. Further distance from home to the treatment centre per 100 km (estimate −3.7, 95% CI −5.7 to −1.9, *p* ≤ 0.001) was significantly associated with lower COST scores, indicating increased financial toxicity. Married adult patients had substantially lower COST scores than single patients (estimate: −9.1, 95% CI −14.8 to −3.4, *p* ≤ 0.001). The median OOP cost was 2050 Swiss francs (CHF) and was spent mainly on travel, accommodation, and eating out. Sixty-three (43%) patients used their savings; 54 (37%) cut spending on leisure activities; 21 (14.4%) cut living expenses; 14 (9.6%) borrowed money; nine (6.2%) worked more; and four (2.7%) sold property. Patients with high COST scores used significantly fewer coping strategies such as saving on leisure activities (estimate −9.5, 95% CI −12.4 to −6.6, *p* ≤ 0.001), spending savings (estimate −3.9, 95% CI −6.3 to −1.4, *p* = 0.002), borrowing money (estimate −6.3, 95% CI −10.4 to −2.2, *p* = 0.003), and increasing workload (estimate −5.5, 95% CI −10.5 to −0.4, *p* = 0.035). Interpretation: A substantial number of cancer patients treated with proton therapy experience financial toxicity in Switzerland. Long travel distances to the proton therapy centre and low income negatively affect the financial well-being of these patients during proton therapy.

## 1. Introduction

Proton therapy is a type of radiation therapy that can precisely deliver a high-radiation dose to the tumour while minimising the irradiation of the surrounding healthy tissue. It plays an important role in managing tumours that are radio-resistant or close to organs at risk of radiation-induced toxicity. This makes it particularly important for paediatric patients, due to their long life expectancy and sensitivity to radiation, or for adults with challenging tumours in the vicinity of critical structures.

The Centre for Proton Therapy at the Paul Scherrer Institute (PSI) in Switzerland is a world leader in this treatment, having fostered the technical development of proton therapy pencil beam scanning and its medical application in cancer patients.

In Switzerland, the Federal Office of Public Health and health insurance providers have agreed to a list of indications for which the cost of proton therapy at PSI will be reimbursed. This includes meningioma, low-grade gliomas, tumours in the base of the skull and head and neck area, sarcomas, chordomas, chondrosarcomas, eye tumours, and all paediatric tumours. Patients from all over Switzerland and abroad are referred to PSI.

Although Swiss insurance providers reimburse the costs of proton therapy, patients may have to pay out to cover the costs of housing and travel during their five to seven weeks of treatment. No data are available on these out-of-pocket (OOP) costs. However, studies from the United States indicate that they are higher among cancer patients than other patient groups [1,2]. High OOP expenses can cause substantial financial distress for the patient and their family [3].

Treatment-related financial distress adversely affects patients’ quality of life, treatment choice, treatment compliance, and treatment outcome [4]. It can be just as toxic as chemotherapy or radiation therapy. In the worst case, it can result in financial insolvency and subsequently contribute to increased patient mortality [5]. Therefore, treatment-related financial distress has been defined as financial toxicity [3].

Reported risk factors for financial toxicity are younger age at cancer diagnosis, lower income, minority race, and loss of employment [6]. Even patients with good insurance coverage, such as those in the Swiss health care system, may suffer from financial toxicity and need familial or societal assistance. Physicians’ engagement with the issue may not meet patients’ expectations. Although many physicians attempt to help patients with financial issues, more than two-thirds of patients have reported that their cancer physician and staff did not substantially increase their financial well-being [7].

Most research on cancer-related financial toxicity has focused on patients from the United States, where there is a fee-for-service system. There is no equivalent research on cancer patients treated with proton therapy in Europe. Our study, based on patient-reported outcomes, aims to capture financial toxicity among cancer patients in Switzerland and explore coping strategies from a patient perspective.

## 2. Material and Methods

### 2.1. Study Design

This is a prospective, observational, cross-sectional study of self-reported financial toxicity (SFT) in cancer patients undergoing proton therapy at PSI. Participants were recruited at the beginning of their treatment and provided written informed consent for participation. OOP costs were documented during the treatment period. A cross-sectional picture of financial toxicity and financial coping strategies was obtained at the end of treatment. The Swiss Ethics Committee reviewed this study protocol and raised no ethical concerns.

### 2.2. Setting

The recruitment period and concurrent data collection spanned from September 2019 to November 2021. All questionnaires used are listed in the Appendix A. After giving informed consent, participants completed a basic questionnaire collecting demographic data (Appendix A). During treatment, participants were asked to keep a cost diary (Appendix A) recording their treatment-related OOP costs. At the end of treatment, participants were invited to fill out the COST questionnaire (Appendix A) [8,9], a questionnaire on financial coping strategies (Appendix A) [10], and to provide information on yearly net household income.

### 2.3. Participants

All cancer patients undergoing proton therapy at PSI were eligible to participate in this study if they spoke German, French, Italian, or English. There were no restrictions regarding diagnosis, comorbidities, or age. In the case of paediatric patients, parents/legal guardians/caregivers were invited to participate. For the statistical analysis, only patients living in Switzerland were included, as the primary aim of this study was to evaluate the financial toxicity of Swiss cancer patients undergoing proton therapy. Data from foreign patients will be later assessed in a separate analysis.

### 2.4. Outcomes and Independent Variables

The primary outcome is an SFT score obtained from a validated questionnaire. Secondary outcomes include the OOP costs and the coping strategies used by patients.

Information on patients and treatment characteristics was collected through the questionnaire and the electronic patient file (ARIA^®^ Oncology Information System, Varian Medical Systems Inc., Palo Alto, CA, USA).

### 2.5. Data Sources/Measurement

Financial toxicity was evaluated and expressed as a score using the COST questionnaire from FACIT, a validated 12-item patient-reported outcome measure (PROM) designed to assess financial toxicity in chronically ill patients (Appendix A) [8,9]. The questionnaire consists of eleven statements on financial situation, concerns about income loss, ability to meet monthly expenses, and other financial problems related to cancer treatment. The twelfth item summarises the overall situation and is not included in the final score calculation. Using a Likert scale from 0 (“Not at all”) to 4 (“Very strongly”), participants indicate the extent to which each statement applies to their experience over the past seven days. Each item is summed up to calculate an overall score of SFT, with some items being reverse-scored. The overall score ranges from 0 to 44, with a lower score indicating low financial well-being.

Because we wanted to include parents and caregivers of children with cancer in this study, we prepared a version of the COST questionnaire for this group under FACITs guidance.

The COST questionnaire was previously only available in English, which is not an official language in Switzerland (although it is often used to overcome language barriers). Therefore, we translated the COST questionnaire into the three prevalent official Swiss languages (German, French, and Italian) with the support and validation of FACIT according to their certified translation method [11].

Coping strategies were assessed using a polar questionnaire based on that used in a study by Zafar et al. [3]. The questionnaire gave six financial coping strategies as options, and participants could manually add further financial coping strategies. In addition, participants were asked to report their annual net household income in Swiss francs (CHF) in the following categories: <45,000, 45,000 to <65,000, 65,000 to <85,000, 85,000 to <105,000, 105,000 to <25,000, 125,000 to <145,000, and ≥145,000. For the statistical analysis, these income categories were condensed into low, medium, and high-income according to the following scheme: <65,000, 65,000 to <125,000, and ≥125,000, respectively. One Swiss franc was valued at an average of €1.0801 for this study period (September 2019 to November 2021).

Participants’ demographic data were collected from the electronic patient file and automatically transferred to this study database. Further information was collected using a basic questionnaire, which participants filled out at the beginning of their treatment, covering nationality, employment status, insurance type and coverage (standard, half-private, or private), living situation, and means of transport during proton therapy (Appendix A). Patients’ home postcodes were matched with Swiss population data from the Federal Statistical Office [12], which classifies each municipality as urban, suburban, or rural.

Each week, participants recorded OOP costs during treatment using a cost diary log. Only treatment-related costs not reimbursed by health insurance were recorded (Appendix A). The diary included six medical and six non-medical subcategories. The medical subcategories were: medication, stationary stays in the hospital, ambulant medical consultations, ambulant therapy, nursing aids, and medical services. The non-medical subcategories were: commute travel/drive to PSI and back; accommodation due to treatment; external meals due to treatment; phone calls due to treatment; household help due to treatment; and leisure activities due to treatment. Participants were also invited to add additional treatment-related costs that did not apply to the categories (free text).

### 2.6. Study Size

For the estimation of the sample size, a precision-based approach was used based on the width of the confidence interval rather than on power [13]. The parameters of the sample size can be justified using a recent study by Huntington et al. [10].

For the primary outcome, the survey of financial toxicity using the COST score, we needed to estimate the COST mean in a Swiss population. The study by Huntington et al. found the COST mean in the United States to be 23.5. We assumed that the mean in the Swiss population would be somewhat higher but that the corresponding standard deviation (SD) would be comparable to that in Huntington’s study (11.1).

The width of a 95% confidence interval for the population mean is roughly four times the standard error (S.E.), and therefore the width of the 95% confidence interval in our study was calculated to be 3.6 [14].

Even with the expected dropout rate of 10% of patients, which would reduce our sample size to 150 − 15 = 135 patients, the width of the corresponding 95% confidence interval would be less than four units and would be acceptably accurate.

### 2.7. Statistical Analysis

The collected data were managed in a database linked to electronic patient files and subsequently anonymised. The characteristics of the participants were summarised using descriptive statistics. The median and interquartile ranges (IQR) were reported for continuous and ordinal variables. Absolute numbers and percentages of the total were calculated for categorical variables.

Mean COST scores were calculated and reported with 95% confidence intervals. Coping strategies were reported in absolute numbers and as a percentage of the total. Furthermore, the use of coping strategies in the adult cohort and caregivers was compared with Chi-squared tests or Fisher’s exact test for the number of observations less than five.

Multiple linear regression models were used to quantify the association between disease-specific and patient-specific parameters and COST outcomes. The parameters were pre-selected based on the study by Huntington et al. [10]. The estimated coefficients were reported with their 95% confidence intervals.

The first regression model included the whole cohort and considered the following independent variables: income category, demographic of hometown, distance to home address, time since diagnosis, and treatment of the first diagnosis versus recurrence or progression. A second regression model was applied to the adult cohort and included variables only applicable to adults, such as age, gender, relationship status, highest educational level, employment category, and income category. The third regression model was applied again to the whole cohort and evaluated the influence of coping strategies on self-reported financial toxicity. This model considers the income category and the six coping strategies assessed in the polar questionnaire (see Appendix A). This was an exploratory analysis to generate future research hypotheses.

All statistical analyses were performed in the R system for statistical computing version 4.2.2 (R Core Team 2022) [15].

## 3. Results

### 3.1. Participants

Between September 2019 and December 2021, 291 patients were treated with proton therapy at PSI. Approximately 264 (72%) who spoke one of the required languages were informed about this study, and 190 (65.3%) individuals agreed to participate. A total of 185 (97.4%) completed the questionnaires, of whom 146 (78.9%) lived in Switzerland and were included in the analysis (Figure 1).

Table 1 summarises the baseline characteristics of patients. The patient population consisted of 90 (61.6%) adults and 56 (38.4%) paediatric patients with median ages of 48.5 years (IQR [37.3; 61.8], range 20–78) and 9.5 years (IQR [5.8; 13.0], range 1–17), respectively. In the adult and paediatric cohorts, 36 (40.0%) and 21 (37.5%) were female.

Significantly more adult patients (31.1%) than caregivers of paediatric patients (12.5%) had an annual household income of <CHF 65,000 (*p* = 0.01).

Approximately 26% of the adult patients and 35.7% of maternal and 25.0% of paternal caregivers of paediatric patients had a university education (*p* = 0.3).

### 3.2. Primary Outcome

The entire cohort had a median COST score of 29.9 (IQR [21.0; 36.0]) (Table 2).

In the multivariable linear regression analysis of the whole cohort, higher income (*p* < 0.0001) and shorter distance to home (*p* = 0.0002) were significantly associated with higher COST scores, in other words, better financial well-being (Table 3).

There was no evidence for an association between the COST score and residence in urban or suburban regions compared to rural areas. There was also no evidence for a difference in whether the diagnosis was initial or a recurrence. The time between diagnosis and the start of proton therapy also had no influence on the COST score (Table 3).

Associations between the adults’ COST scores and patients’ characteristics were analysed in a second multivariable regression model. There was strong evidence for being married or in a permanent partnership to be associated with a lower COST score than being single or widowed (married *p* = 0.002, partnership *p* = 0.008 compared to single or widowed) (Table 4). In this model, there was no evidence that age, patient gender, highest educational level, and employment status were associated with COST scores.

A further linear regression analysis of the COST score was performed using treatment site, concurrent chemotherapy, and treatment duration as explanatory variables. There was no statistical association between these variables and financial toxicity (see Appendix A).

### 3.3. Coping Strategies

Of the 146 participants, 54 (37.0%) reported that they had to reduce their spending on leisure activities due to additional expenses related to proton therapy treatment. Among them, there was weak evidence that caregivers of paediatric patients (48.2%) were more affected than adult patients (30.0%) (*p* = 0.041). Approximately 43% of patients used their savings, 9.6% had to borrow money, 14.4% had to reduce their spending on food and clothing, and 2.7% had to sell property (Table 5).

There was no evidence for a difference between the adult cohort and the caregivers for the other coping strategies. Linear regression analysis showed that saving on leisure activities (*p* < 0.0001), spending savings (*p* = 0.002), borrowing money (*p* = 0.003), and working more (*p* = 0.035) were associated with a lower COST score (Table 6).

### 3.4. Out-of-Pocket Costs

A total of 139 (96%) patients filled out the cost diary.

The median total OOP cost during the treatment period was CHF 2050.0 (IQR [759.9; 4434.0]). Most OOP expenses went on travel to and from the proton therapy centre (median: CHF 800 (IQR [367.3; 1843.0]). Eating out was the second highest expenditure category (median CHF 200.0, IQR [0.0; 533.0]). None of the patients had additional expenses for medical items not reimbursed by health insurance.

Figure 2 shows the total OOP costs, total non-medical OOP costs, prices for travelling to and from PSI during treatment, and costs for external meals separated by income category.

The OOP cost for patients in the lowest income class was CHF 2640.0 [IQR 779.7; 4642.0]. In the middle-income category, the median OOP cost was CHF 1866.0 [IQR 700.3; 5154.5]; in the highest-income class, it was CHF 2110.0 [IQR 1162.0; 3823.0].

Patients in the lowest income category spent the most on travel expenses (median CHF 890.0, [IQR 540.0; 1782.0]). In contrast, patients in the middle-income category spent a median of CHF 772.5 (IQR [361.4; 1887.5]) on travel expenses, and patients in the high-income category spent a median of CHF 709.0 (IQR [326.0; 1786.0]).

## 4. Discussion

This study is the first on financial toxicity among patients receiving proton therapy in Switzerland. It is also the first study on financial toxicity to include caregivers of paediatric cancer patients.

The findings demonstrate that patients treated with proton therapy in Switzerland face acute financial toxicity. Half of the participants reported having to use their savings to meet additional expenses related to receiving proton therapy, and a third cut down on leisure activities. In addition, 14% had to cut back on basic needs such as food and clothing, and nearly 10% had to borrow money.

For most coping strategies, there was no significant difference between the adult cohort and the caregivers of paediatric patients. However, caregivers of children with cancer spent significantly less on leisure activities than the adult cohort (*p* = 0.04).

Patients paid, on average, CHF 2050 for non-medical OOP costs during treatment. This corresponds to about 1.9% of the Swiss median net household income of CHF 109,760 in 2022 [16,17].

Interestingly, Switzerland is one of the OECD countries where the proportion of medical OOP costs out of total household expenditure is the highest [18]. Health insurance in Switzerland is compulsory and is financed by the contributions of the insured (premiums), the cost-sharing of the insured (deductible), and federal and cantonal funds. Under basic insurance, all insured people have the same benefits, and sick people can receive their salary for up to two years. In principle, neither the patient nor the caregivers should be financially burdened. Nevertheless, health insurance does not fully cover many of the costs related to treatment.

Our study found that the largest share of non-medical OOP costs was for daily travel costs, which amounted to a median of CHF 800 (39% of the median OOP expenses) for an average duration of proton therapy of about six weeks. Added to this are the costs of buying food when away from home, often in restaurants, and other non-medical needs such as childcare and household help.

We did not observe evidence for a difference in the level of OOP costs in the different income categories. However, evidence for an association was observed between COST scores and the distance between the patient’s postal address code and the proton therapy centre. Patients living further away from the treatment centre face higher travel costs and are at a higher risk of financial toxicity.

The study by Huntington et al. showed no significant association between distance from zip code to facility and COST score [10]. However, only 38% of patients lived more than 64 km (>40 miles) away from the treatment centre in the Huntington cohort, whereas in our cohort the median distance from home was 83 km (52 miles). This may be because PSI is the only institution offering proton therapy in Switzerland and is located in a decentralised rural area. As a result, we found strong evidence that patients living further away from treatment facilities face higher financial toxicity than patients living closer.

Other papers do align with our study’s finding that financial toxicity is high among patients who have to travel a long distance for treatment [19]. Travel costs play a particularly major role in radiation therapy because patients have to come for treatment every day for five to seven weeks. The effect is pronounced in proton therapy, where the number of centres is limited and the catchment area is large [20,21]. This leads to unequal access to proton therapy, as recently shown for paediatric patients with primary central nervous system (CNS) tumours and solid malignancies [21,22]. A similar disparity in access to proton therapy has been found in prostate cancer patients [23]. Remoteness and related financial toxicity decrease the likelihood of cancer patients receiving proton therapy.

This clearly raises ethical issues about equitable access to cancer treatment. Utmost care should be taken in the geographical planning of proton therapy centres to ensure that travel costs are kept to a minimum for patients and caregivers alike.

Our analysis of net annual household income showed that those on middle incomes made up half our cohort (48.7%), corresponding to the Swiss median household equivalent income of CHF 109,760 in 2022 [16,17].

It is worth noting that a substantial number of patients in our cohort were below this national median and were therefore at risk of financial toxicity (estimate 8.8 patient/caregiver income < 64,000).

Not surprisingly, an association was found between an annual household net income of CHF ≥ 125,000 and a high COST score. Conversely, there was strong evidence of financial toxicity in patients with an annual net income of <64,000 (estimate 8.8). The same finding is consistently reflected in other financial toxicity studies [9,10,24,25,26,27,28], demonstrating that low household income is one of the most important factors associated with higher financial toxicity. This can sometimes influence the selection of treatment [22].

Socio-demographic factors such as age, race, family status, educational status, and geographical origin also play an important role. A systematic review of studies worldwide found evidence that cancer survivors experience financial hardship regardless of the measurement method used, and the main risk factors for financial toxicity are female gender, younger age, low income, adjuvant therapies, and recent diagnosis [29].

However, unlike other studies [10,24], ours did not find evidence for age-affecting financial toxicity, although there was a trend for younger patients to have higher toxicity. As our sample consisted of adult patients and children represented by their caregivers, we could only examine the influence of age in the smaller adult sample (*n* = 90).

We did not find an association between gender and COST score among adults. However, we did observe a tendency toward male patients having higher COST scores (estimate 3.4).

One intriguing aspect of our study was the significant correlation between COST score and relationship status in the adult cohort. Being married or in a permanent partnership, as opposed to being single or widowed, was strongly correlated with a lower COST score, indicating higher financial toxicity (estimate −9). This is in contrast to the Huntington et al. study, where being married or in a stable partnership was found to be a protective factor for financial toxicity [10]. A similar result to Huntington et al. was found in studies by Honda et al. and Offodile et al. [27,30].

This result is challenging to understand at first glance. One possible explanation could be that patients who live alone are less concerned about their responsibility for the financial needs of others. Another possibility to consider is that there might be a selection bias at play. For instance, single patients who belong to the middle or higher-income group without children may have easier access to proton therapy compared to married patients with children. Further investigation is needed since we did not gather information on the parental status of our adult cohort, which could have impacted their finances. It can also be hypothesised that couples in Switzerland have a higher taxation rate on the federal level when compared to single citizens. Couples, especially those with children, have higher non-treatment-related expenses in a country that provides no financial support for day care and no substantial tax reduction for the couple’s children. Of course, this is only a hypothesis that should be further investigated in future studies.

Overall, the results of this study are highly relevant for oncology care providers, specifically those administering proton therapy. Though many providers believe they provide good support services for patients, studies have shown that patients would like greater physician involvement in managing their financial burden. Raising awareness and implementing practical measures to navigate financial toxicity is essential [31].

There were several limitations to this study. First, the results of the multivariable analyses did not generate causal relationships and must be recognised as only hypothesis-generating. Further research is needed to elicit possible causal associations.

Second, this study was conducted during the COVID-19 pandemic, and this may have influenced the results. With national COVID-19 restrictions introduced in March 2020, a large part of our cohort was subject to lockdown measures. Some patients reported choosing to travel in their private car instead of public transport due to the fear of contracting COVID-19. Furthermore, eating out and many leisure activities were not possible during the lockdown, leading to a potential bias in the use of coping strategies and OOP payments. We also did not capture if adult patients in our cohort had children, which could impact their financial well-being.

Third, several selection biases may be present, which may have led to an underestimation of the extent of financial toxicity and excluded patients who were at risk.

Language barriers may be one selection bias, as nine patients with primary residence in Switzerland, including five migrants and four refugees, could not participate because they did not speak one of the required languages.

In addition, many people consider financial problems a sensitive and taboo topic. This is reflected in the fact that approximately 25% of potential study participants decided not to participate. Half of them did not provide an explanation, while the other half cited reasons such as feeling overwhelmed, having no time, or other concerns. This suggests that patients may be hesitant to discuss financial issues. Further investigations are needed to support this hypothesis.

Another selection bias could occur even before the patient is referred for proton therapy, as some patients in the low-income category are already discouraged from considering proton therapy at this stage due to the financial burden, even though it could be of benefit to them.

Even given these limitations, however, the analysis was of a good-sized cohort, which was determined before this study began. To the best of our knowledge, this is the first prospective study worldwide reporting financial toxicity in patients being treated with proton therapy.

Finally, based on the results of this study, the PSIs Centre for Proton Therapy became more aware of the problem of financial toxicity. Severely affected patients can apply for financial support for travel and accommodation costs and receive a financial contribution from a specially-created donation fund after a positive review by an internal committee.

## 5. Conclusions

In summary, in our patient cohort, we showed that financial toxicity primarily impacts low-income patients who reside far from the Swiss proton therapy center. Significantly, in a country with a compulsory health system that fully reimburses proton therapy, a substantial number of patients undergoing this therapy did have substantial OOP costs, which sometimes required them to use their savings, borrow more, or even sell property. Relationship status was another important factor, with financial well-being more associated with single people. In the future, it is crucial to address the problem of financial toxicity at an early stage to mitigate the risks and offer supportive measures.

## Figures and Tables

**Figure 1 cancers-15-05498-f001:**
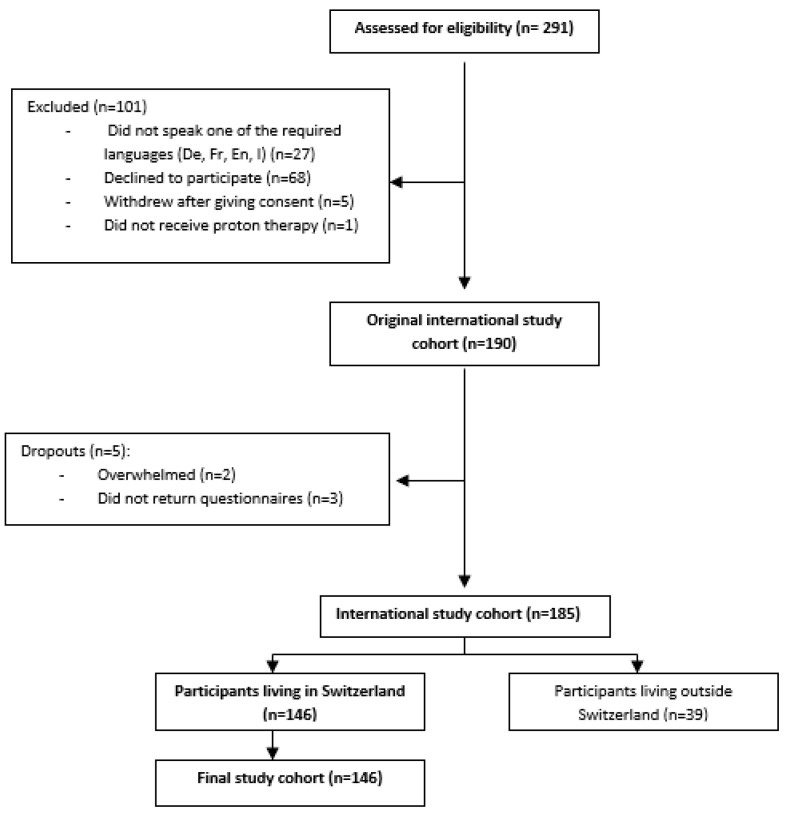
Flow diagram of patient inclusion.

**Figure 2 cancers-15-05498-f002:**
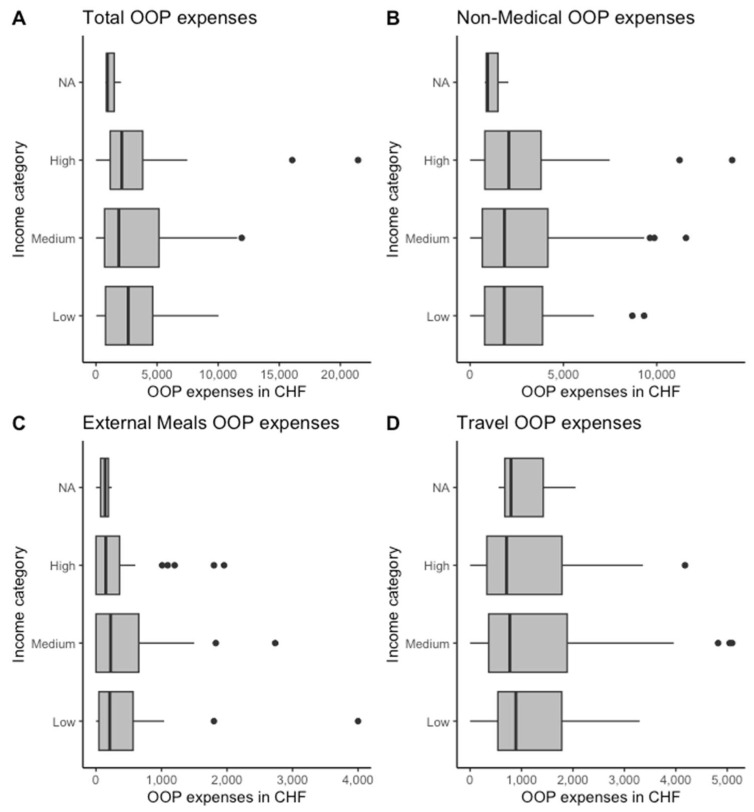
Out-of-pocket (OOP) expenses by income category in total (**A**), for travel (**B**), food (**C**), and non-medical items (**D**).

**Table 1 cancers-15-05498-t001:** Baseline characteristics of patients.

Demographics	Cohort (*n* = 146)	Adults (*n* = 90)	Child Caregivers (*n* = 56)	
Age, years (median, IQR)	30.5 [11.0; 52.0]	48.5 [37.3; 61.8]	9.5 [5.8; 13.0]	
Gender				
- Female	57 (39.0%)	36 (40.0%)	21 (37.5%)	
- Male	89 (61.0%)	54 (60.0%)	35 (62.5%)	
Marital status (*n*, %)				
- Single or widowed		24 (26.7)		
- Permanent partnership		13 (14.4)		
- Married		47 (52.2)		
- Not available		6 (6.7)		
Net household income per year (Swiss francs, CHF)				
Low (<65 kCHF)	35(24.0%)	28 (31.1%)	7 (12.5%)	
Medium (65–<125 kCHF)	71 (48.6%)	37 (41.1%)	34 (60.7%)	
High (≥125 kCHF)	37 (25.3%)	22 (24.5%)	15 (26.8%)	
not reported	3 (2.1%)	3 (3.3%)	0 (0.0%)	
Level of Education			M = maternal P = paternal	
Secondary school		14 (15.6%)	M: 4 (7.1%)	P: 4 (7.1%)
Vocational training		43 (47.8%)	M: 26 (46.4%)	P: 29 (51.8%)
High school		9 (10.0%)	M: 5 (8.9%)	P: 8 (14.3%)
University		24 (26.7%)	M: 20 (35.7%)	P: 14 (25.0%)
Not reported		0 (0.0%)	M: 1 (1.8%)	P: 1 (1.8%)
Employment status			M = maternal, P = paternal	
Employed		53 (58.9%)	M: 39 (69.6%)	P: 45 (80.4%)
Self-reliant		5 (5.6%)	M: 4 (7.1%)	P: 6 (10.7%)
Not gainfully employed		8 (8.9%)	M: 9 (16.1%)	P: 0 (0.0%)
Unemployed		3 (3.4%)	M: 3 (5.4%)	P: 3 (5.4%)
Sickness benefit recipient		1 (1.1%)	M: 0 (0.0%)	P: 1 (1.8%)
Retired		20 (22.2%)	M: 0 (0.0%)	F: 0 (0.0%)
Not reported		0 (0.0%)	M: 1 (1.8%)	F: 1 (1.8%)
Language region				
German-speaking	112 (76.7%)	72 (80.0%)	40 (71.4%)	
French-speaking	25 (17.1%)	13 (14.4%)	12 (21.4%)	
Italian-speaking	9 (6.2%)	5 (5.6%)	4 (7.1%)	
Place of residence				
Urban region	91 (62.3%)	59 (65.6%)	32 (57.1%)	
Suburban region	24 (16.4%)	9 (10.0%)	15 (26.8%)	
Rural region	31 (21.2%)	22 (24.4%)	9 (16.1%)	
Distance from home (km) to proton therapy centre, median (IQR)	83 (52; 153.0)	83.5 (55.0; 147.5)	82 (46.5; 187.0)	
Diagnosis				
Primary cancer site				
- Brain and skull base	86 (58.9%)	48 (53.3%)	38 (67.9%)	
- Head and neck	24 (16.4%)	21 (23.3%)	3 (5.4%)	
- Extracranial	36 (24.7%)	21 (23.3%)	15 (26.8%)	
Recurrent tumour, *n* (%)	25 (17.1%)	19 (21.1%)	6 (10.7%)	
Time since diagnosis to proton therapy months (IQR)	2.9 (1.7; 4.8)	3.0 (1.9; 6.7)	2.9 (1.5; 4.4)	
Treatment				
Duration of proton therapy (days), median (IQR)	43 (41; 49)	44 (42; 51),	43 (37; 44)	
Proton dose Gy (RBE), median (IQR)	54 (54; 65)	60 (54;70)	54 (50.4; 54.7)	
Chemotherapy concomitant to proton therapy, *n* (%)	29 (19.9%)	9 (10.0%)	20 (35.7%)	

**Table 2 cancers-15-05498-t002:** COST score: a lower score indicates low financial well-being.

COST Score	Median	IQR
Full cohort (*n* = 146)	29.9	21.0, 36.0
Adults (*n* = 90)	30.0	21.3, 37.9
Female adults	29.0	21.0, 34.5
Male adults	32.0	22.5, 38.9
Paediatric patients (<18 years) (*n* = 56)	28.0	20.5, 34.0

**Table 3 cancers-15-05498-t003:** Linear regression analysis of the COST score of the whole cohort (*n* = 146), including income, area of residence, distance from home to proton therapy centre (per 100 km), time since diagnosis (months), and recurrent tumour status as explanatory variables.

		Estimate	Lower 95% CI	Upper 95% CI	*p*-Value
Intercept		27.98	22.8	33.2	<0.0001
Income					
	Medium vs. low	2.8	−1.0	6.6	0.15
	High vs. low	8.1	3.7	12.4	0.0003
Residence					
	Urban vs. rural	0.7	−3.2	4.6	0.72
	Suburban vs. rural	2.1	−2.9	7.2	0.40
Distance from home to the proton centre (per 100 km)		−3.8	−5.69	−1.9	0.0002
Time since diagnosis (months)		0.1	−0.2	0.5	0.44
Treatment of recurrent tumour or progression vs. first diagnosis		−1.6	−5.7	2.5	0.44

**Table 4 cancers-15-05498-t004:** Linear regression analysis of the COST score of the adult cohort (*n* = 90), including age, gender, relationship status, education, and employment status as explanatory variables.

		Estimate	Lower 95% CI	Upper 95% CI	*p*-Value
Intercept		22.5	12.08	32.93	<0.0001
Male vs. female		3.4	−1.22	8.03	0.15
Relationship status					
	Married vs. single	−9.1	−14.8	−3.4	0.002
	Permanent partnership vs. single	−9.4	−16.2	−2.6	0.008
Educational level					
	Occupational training vs. secondary school	2.3	−3.8	8.4	0.46
	High school vs. secondary school	4.3	−4.6	13.3	0.34
	College, university vs. secondary school	3.8	−2.9	10.6	0.26
Employment					
	Unemployed vs. employed	−0.0	−6.7	6.6	0.99
	Retired vs. employed	−2.0	−9.5	5.6	0.60
Age (in years)		0.2	−0.0	0.4	0.08

**Table 5 cancers-15-05498-t005:** Patients’ self-reported coping strategies during proton therapy for the whole cohort (*n* = 146).

Coping Strategies	Whole Cohort (*n* = 146)		Adult (*n*= 90)		Caregivers (*n* = 56)		*p*-Value
	*n*	%	*n*	%	*n*	%	
I/we need to reduce our spending on leisure activities.	54	37.0	27	30.0	27	48.2	0.041
I/we need to reduce our spending on the basics of life.	21	14.4	12	13.3	9	16.1	0.83
I/we need to use our savings or part of the savings.	62	42.5	33	36.7	29	51.8	0.10
I/we need to borrow money.	14	9.6	6	6.7	8	14.3	0.22
I/we need to sell our property or part of it.	4	2.7	3	3.3	1	1.8	1
My partner or I need to work more.	9	6.2	5	5.6	4	7.1	0.73

**Table 6 cancers-15-05498-t006:** Linear regression analysis of the COST score for the whole cohort, including coping strategies as explanatory variables.

		Estimate	Lower 95% CI	Upper 95% CI	*p*-Value
Intercept		34.06	31.38	36.73	<0.0001
Income					
	Medium vs. low	0.8	−2. 00	3.5	0.56
	High vs. low	0.9	−2.3	4.2	0.58
Coping strategies					
	Spending less on leisure activities	−9.5	−12.4	−6.6	<0.0001
	Spending less on basics	−0.7	−4.7	3.2	0.72
	Spending savings	−3.9	−6.3	−1.4	0.002
	Borrowing money	−6.3	−10.4	−2.2	0.003
	Selling property	−6.5	−13.3	0.4	0.063
	Working more	−5.5	−10.5	−0.4	0.035

## Data Availability

The data presented in this study are available in this article.

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
