# Peer review of "Financial Toxicity in Swiss Cancer Patients Treated with Proton Therapy: An Observational Cross-Sectional Study on Self-Reported Outcome"

_cancers, 2023, doi:10.3390/cancers15235498_

Round 1

Reviewer 1 Report

Comments and Suggestions for Authors

This is a well written paper on financial toxicity in cancer patients receiving proton therapy.  The intro, methods, and results are clear and well written.  I do feel that more discussion about limitations and its impact on recommendations could be added.   Included in this is the potential effect of selection bias (those unable to afford treatment/do not seek treatment and therefore not included in this study).  As a consequence it is possible this sample is NOT representative of the full eligible population.   I have provided further details in my attached marked up PDF document.

Comments on the Quality of English Language

Well written with only a few minor suggestions in attached document.

Reviewer 2 Report

Comments and Suggestions for Authors

Thanks for inviting to review this paper. I think the paper is good. I suggest for the authors to add other variables available to them for mulvar. analysis

my comments:

1. abstract - estimates -- add 95CI

2. M&M - a large number of patients declined to participate. is this systematic? please comment and analyse if possible.

3. significant number also don't speak the languages - are they immigrants? comment. this may cause systematic exclusion.

4. Please reanalyse with site of tx, chemo and treatment duration as explanatory variables

Round 2

Reviewer 1 Report

Comments and Suggestions for Authors

I am satisfied with the responses to my questions/concerns.  I have no further comments.

Reviewer 2 Report

Comments and Suggestions for Authors

Accept